# Correction of amblyopia in cats and mice after the critical period

Ming-fai Fong[1†], Kevin R Duffy[2†], Madison P Leet[1], Christian T Candler[1], Mark F Bear[1]*

[1]The Picower Institute for Learning and Memory, Department of Brain and Cognitive Sciences, Massachusetts Institute of Technology, Cambridge, United States; [2]Department of Psychology and Neuroscience, Dalhousie University, Halifax, Canada

**Abstract** Monocular deprivation early in development causes amblyopia, a severe visual impairment. Prognosis is poor if therapy is initiated after an early critical period. However, clinical observations have shown that recovery from amblyopia can occur later in life when the non-deprived (fellow) eye is removed. The traditional interpretation of this finding is that vision is improved simply by the elimination of interocular suppression in primary visual cortex, revealing responses to previously subthreshold input. However, an alternative explanation is that silencing activity in the fellow eye establishes conditions in visual cortex that enable the weak connections from the amblyopic eye to gain strength, in which case the recovery would persist even if vision is restored in the fellow eye. Consistent with this idea, we show here in cats and mice that temporary inactivation of the fellow eye is sufficient to promote a full and enduring recovery from amblyopia at ages when conventional treatments fail. Thus, connections serving the amblyopic eye are capable of substantial plasticity beyond the critical period, and this potential is unleashed by reversibly silencing the fellow eye.

**\*For correspondence:**
mbear@mit.edu

†These authors contributed equally to this work

**Competing interest:** The authors declare that no competing interests exist.

## Introduction

Amblyopia is a prevalent form of visual disability that emerges during infancy when inputs to the visual cortex from the two eyes are poorly balanced (*Simons, 2005*). The most common causes of amblyopia are strabismus and asymmetric refraction, but the most severe type—deprivation amblyopia—arises from opacities or obstructions of form vision (e.g., by cataract). The current standard of care is to restore clarity (e.g., by cataract extraction) and focus, and then promote recovery of the weak amblyopic eye by temporarily patching the fellow eye (*Wallace et al., 2018*). However, the effectiveness of occlusion therapy is limited by poor compliance, variable recovery outcomes, and a significant risk of recurrence. Additionally, occlusion therapy is largely ineffective if it is initiated after age 10 (*DeSantis, 2014*) or, in the case of deprivation amblyopia caused by congenital cataract, after the first year of life (*Birch and Stager, 1996*). The need for improved treatments for amblyopia is widely acknowledged (*Falcone et al., 2021*; *Quinlan and Lukasiewicz, 2018*).

Studies over many decades in cats and monkeys have shown how temporary monocular deprivation (MD) sets in motion a series of changes in primary visual cortex (V1) that degrade vision through the deprived eye. As in humans, these changes can be reversed by temporarily occluding the fellow eye, but the effectiveness of this procedure is again limited to a brief critical period (*Blakemore et al., 1978*; *Blakemore and Van Sluyters, 1974*). Rodents have become the dominant animal model for study of the synaptic basis of amblyopia, and an important recent development has been the finding from multiple laboratories that diverse manipulations, all culminating in a global reduction in inhibition by a population of cortical interneurons, can allow recovery from the effects of MD at ages beyond the classically defined critical period (*Bavelier et al., 2010*; *Hensch and Quinlan, 2018*; *Sengpiel, 2014*). As exciting as these results are, however, most of the manipulations used in mice and rats to promote adult recovery have limited applicability to humans. Some are not feasible in a therapeutic

setting (e.g., interneuron transplantation [*Davis et al., 2015*]) and others require systemic exposure to agents that have actions beyond visual cortex (e.g., cholinesterase inhibitors [*Morishita et al., 2010*] and ketamine [*Grieco et al., 2020*]), and none have demonstrated meaningful clinical efficacy in a human study to date (e.g., fluoxetine [*Huttunen et al., 2018*], citalopram [*Lagas et al., 2019*], levodopa [*Repka et al., 2015*], donepezil [*Chung et al., 2017*]). Indeed, some treatments that work robustly in rodents have yielded disappointing results when investigated in other species with a more differentiated visual system, like the cat (e.g., *Holman et al., 2018*; *Vorobyov et al., 2013*). It is now recognized that studies limited to a single species, particularly rats or mice with primitive visual systems, may be unreliable guides to human amblyopia treatment (*Mitchell and Sengpiel, 2018*).

In the current study we took a 'bedside to bench' perspective. One interesting observation in the human clinical literature is that significant recovery from amblyopia can sometimes occur in adults when the normal (fellow) eye is damaged or removed (enucleated) following injury or disease (*El Mallah et al., 2000*; *Kaarniranta and Kontkanen, 2003*; *Klaeger-Manzanell et al., 1994*; *Rahi et al., 2002*; *Vereecken and Brabant, 1984*). Similar findings in cat and monkey models have been interpreted to mean that a mechanism contributing to amblyopia is the continuous suppression of responses to the deprived eye by activity through the fellow eye that is relieved by enucleation (*Harwerth et al., 1984*; *Hendrickson et al., 1977*; *Hoffmann and Lippert, 1982*; *Kratz and Spear, 1976*). According to this view, fellow eye enucleation enables detection of previously subthreshold input from the amblyopic eye simply by decreasing inhibitory tone in V1, similar to what is observed following direct pharmacological blockade of GABAergic inhibition with bicuculline (*Burchfiel and Duffy, 1981*; *Duffy et al., 1976*; *Sillito et al., 1981*). If this were the sole basis for the observed recovery, amblyopic eye responses would again disappear if inhibition was restored (*Bear et al., 1985*). An alternative but not unrelated hypothesis is that eliminating activity in the strong eye via enucleation allows experience through the amblyopic eye to drive long-lasting synaptic strengthening. A window of opportunity for recovery could be opened by even the temporary reduction in inhibition, as well as by a number of other homeostatic molecular mechanisms that promote excitatory synaptic plasticity (*Cho and Bear, 2010*; *Cooper and Bear, 2012*; *Lee and Kirkwood, 2019*; *Li et al., 2019*). If this idea is correct, merely silencing the fellow eye temporarily should be sufficient to enable a lasting recovery. Thus, the 'suppression removal' hypothesis predicts that amblyopia will be immediately mitigated by silencing the fellow eye, but only for as long as that eye is inactive, whereas the 'synaptic strengthening' hypothesis predicts that recovery will outlast the period of retinal inactivation. We have conducted experiments in two species, mouse and cat, to distinguish among these alternatives. Our results show that temporary inactivation of the fellow eye after long-term MD can permanently restore vision through both eyes.

## Results

### Temporary monocular inactivation potentiates non-inactivated eye responses in mouse V1

We first sought to characterize the impact of temporary retinal inactivation on the cortical responses to visual stimulation in neurotypical mice, initiated at postnatal day (P) 47, which is after the critical period has ended (*Gordon and Stryker, 1996*). Recording electrodes were implanted into the binocular zone of V1 layer 4 to measure baseline visual evoked potentials (VEPs) in awake, head-fixed animals (*Figure 1A–B*; *Figure 1—figure supplement 1*). In agreement with previous studies, under baseline conditions, the response to stimulation of the contralateral eye was approximately double that of the ipsilateral eye (*Figure 1C–D*). The contralateral retina was then inactivated for 1–2 days using the voltage-gated sodium channel blocker, tetrodotoxin (TTX), injected into the vitreous humor. During the period of inactivation, responses to stimulation of the contralateral eye predictably fell to noise levels (*Figure 1C*). Meanwhile, responses to stimulation of the non-inactivated ipsilateral eye rose dramatically, as would be expected by loss of interocular suppression (*Figure 1D*; *Pietrasanta et al., 2014*). We continued to track contralateral and ipsilateral VEPs for several days as the effect of TTX dissipated and retinal activity returned. Importantly, responses measured during stimulation of the contralateral eye returned to baseline levels (*Figure 1C and E*), indicating that retinal inactivation was fully reversible. Non-inactivated (ipsilateral) eye responses remained elevated above baseline after the contralateral eye responses returned, but diminished significantly over the

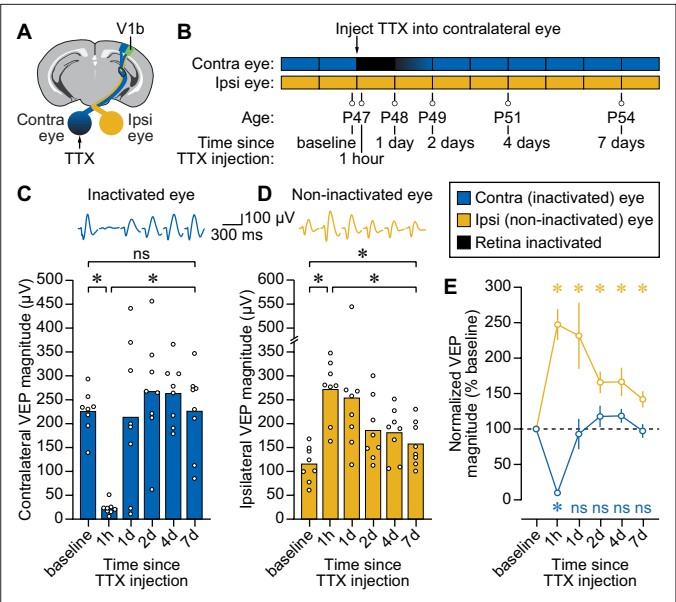

**Figure 1.** Temporary inactivation of one retina potentiates visual responses to stimulation of the non-inactivated eye in awake, late-adolescent mice. (**A**) Schematic of mouse brain showing recording site in binocular V1 (V1b) and tetrodotoxin (TTX) injection site into contralateral eye. (**B**) Experimental timeline showing ocular manipulation and visual evoked potential (VEP) recording session times. (**C**) Longitudinal measurements of VEP magnitude for stimulation of the contralateral, inactivated eye. Filled bars denote mean peak-to-peak magnitude and open circles denote individual biological replicates. Average VEP waveforms are shown above for each time point. Asterisk denotes a statistically significant difference (p < 0.05) and ns denotes p > 0.05. Analyses were performed using one-way repeated measures ANOVA (F = 12.95, p = 0.0013). For comparisons of interest, Sidak's post hoc tests were performed with correction for multiple comparisons (baseline vs. 1 hr, p < 0.0001; 1 hr vs. 7 days, p = 0.0008; baseline vs. 7 days, p > 0.9999). (**D**) Same as C for non-inactivated, ipsilateral eye. Analyses were performed using one-way repeated measures ANOVA (F = 9.361, p = 0.0092) followed by Sidak's post hoc tests (baseline vs. 1 hr, p < 0.0001; 1 hr vs. 7 days, p < 0.0001; baseline vs. 7 days, p = 0.0121). (**E**) VEP magnitude over time normalized to baseline. Blue trace denotes inactivated contralateral eye; yellow trace denotes non-inactivated ipsilateral eye; open circles denote mean; error bars denote SEM; blue and yellow asterisks denote significant differences (p < 0.05) compared to hypothesized value of 100 % for the contralateral and ipsilateral eyes, respectively (ns denotes p > 0.05). Analyses were performed using a one-sample t-test for the contralateral eye (compared to baseline: 1 hr, p < 0.0001; 1 day, p = 0.7475; 2 days, p = 0.2739; 2 days, p = 0.0976; 7 days, p = 0.7856) and a one-sample Wilcoxon test for the ipsilateral eye (compared to baseline: 1 hr, p = 0.0078; 1 day, p = 0.0078; 2 days, p = 0.0156; 2 days, p = 0.0156; 7 days, p = 0.0156), with test identity selected based on outcome of Shapiro-Wilk normality test. All data in this figure is for phase-reversing sinusoidal grating stimulation at a spatial frequency of 0.2 cycles per degree (cpd).

The online version of this article includes the following source data and figure supplement(s) for figure 1:

**Source data 1.** VEP magnitudes in mouse V1 following monocular retinal inactivation.

**Figure supplement 1.** Histological verification of electrode position in mouse binocular primary visual cortex (V1).

course of several days (*Figure 1D–E*). These results demonstrate that removal of the influence of the dominant eye immediately augments responses through the non-dominant eye, consistent with relief from interocular suppression, but can also set the stage for lasting potentiation of responses to the non-inactivated eye.

## Fellow eye inactivation promotes stable recovery following long-termMD in mouse V1

We next asked how temporary inactivation of the fellow eye would impact vision in amblyopic animals. To model deprivation amblyopia, we subjected mice to long-term MD initiated during the peak of the classical critical period for juvenile ocular dominance plasticity and extending 2 weeks beyond its closure

(P26-47). To assess the consequences of fellow eye inactivation in amblyopic mice, we monitored VEPs from V1 just after opening the monocularly deprived eye, and then for several weeks after delivering a single intravitreal injection of TTX into the non-deprived eye (MD then TTX group; *Figure 2A*). Littermate controls were distributed into two additional groups: one undergoing 3 weeks of MD followed by fellow eye saline injection (MD group) and another undergoing sham eyelid suture/re-opening followed by a fellow eye saline injection (Sham group). In all cases, the deprived (or sham deprived) eye was contralateral to the recording electrode while the injected non-deprived fellow eye was ipsilateral. Immediately following eye opening, both groups that underwent MD showed clear depression of deprived contralateral eye VEPs, with response magnitudes approximately half of those observed in sham controls (*Figure 2B–C*). Longitudinal tracking of these animals revealed little change in contralateral visual responsiveness over several weeks in sham or MD animals receiving ipsilateral eye saline injections, confirming stability of the visual deficit. Strikingly, however, animals receiving fellow eye TTX injections following MD showed a significant increase in deprived eye responses (*Figure 2B*) to values that were comparable to sham controls (*Figure 2C*). The recovery of contralateral eye responses to normal levels was stable for many weeks (*Figure 2B–C*) and occurred across a range of spatial frequencies (*Figure 2D*; *Supplementary file 1A*). These results indicate that temporary inactivation of the fellow eye promotes rapid, complete, and apparently permanent recovery of amblyopic eye responses in V1.

We also measured V1 responses to stimulation of the non-deprived fellow eye (*Supplementary file 1B*). Immediately after opening the deprived eye, response magnitudes through the non-deprived fellow eye were slightly elevated above those of sham animals (*Figure 2E–G*), consistent with potentiation of non-deprived eye responses that has been well documented in rodent V1 (*Frenkel and Bear, 2004*). The initial potentiation of fellow eye responses returned to sham levels during the weeks after re-opening the deprived eye (*Figure 2E–G*). These results show that unlike the stable response depression observed for the deprived eye following MD alone (*Figure 2B–D*), potentiation of fellow eye responses following MD appears transient. In addition, the trajectory of fellow eye responses over time was not affected by temporary retinal inactivation of this eye; responses recovered fully and were indistinguishable from sham controls during the weeks after re-opening the deprived eye irrespective of treatment (*Figure 2E–G*). Together, these results demonstrate that following amblyogenic rearing in mice, fellow eye inactivation initiated beyond the juvenile sensitive period fosters a return of normal vision through both eyes.

## Reverse occlusion promotes transient recovery following long-term MD in mouse V1

The standard treatment for amblyopia in infants and young children is to temporarily occlude vision through the fellow eye to promote vision through the amblyopic one. In animal models, this can be simulated using reverse occlusion (RO), wherein the fellow eye is temporarily sutured closed after the period of MD. RO degrades visual responses to patterned stimuli but does not eliminate ganglion cell activity.

To directly compare the impact of fellow eye inactivation and RO on the amblyopic visual cortex, we subjected littermate mice to long-term MD and then either briefly inactivated or sutured closed the fellow eye for 1 week (*Figure 3A*). We controlled for the effect of injection and eyelid suture using saline injections in the RO group and sham eyelid closure/opening in the TTX group. These experiments replicated the finding that a single injection of TTX into the fellow eye promoted recovery of deprived eye responses that was stable for many weeks across a range of spatial frequencies (*Figure 3B–D*; *Supplementary file 2A*). Interestingly, 1 week of RO also led to potentiated responses through the originally deprived eye. However, unlike fellow eye inactivation, the gains observed after RO were short-lived and dissipated over time, presumably because of the age at which it was initiated (P47). Responses to fellow eye stimulation were again elevated immediately after opening the deprived eye, and lessened in the weeks that followed (*Figure 3E–G*; *Supplementary file 2B*). There was no significant difference in fellow eye response magnitude between treatment groups. Collectively, these results indicate that the plasticity driven by RO in adolescent mice is temporary, whereas the recovery driven by fellow eye inactivation is long-lasting.

## Fellow eye inactivation promotes stable recovery following long-term MDin cat V1

We next evaluated fellow eye inactivation as a potential amblyopia treatment in the cat, the species that laid the foundation for much of what is known about ocular dominance plasticity and the

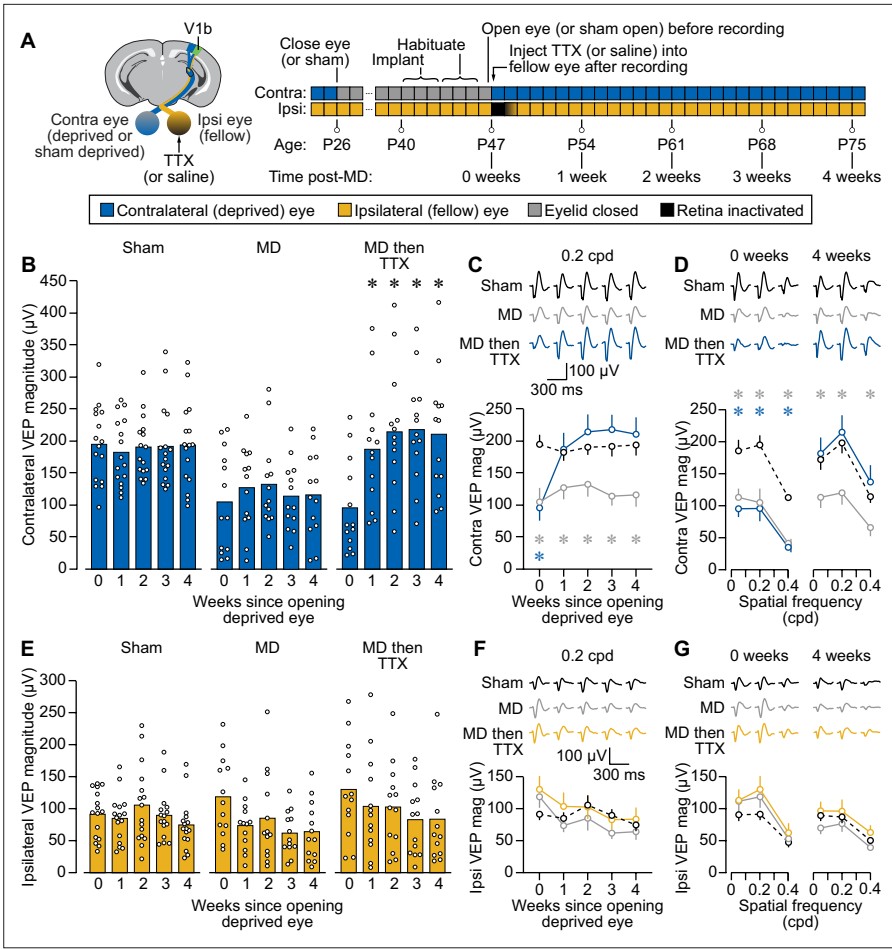

**Figure 2.** Fellow eye inactivation in mice promotes stable and complete recovery of vision in both eyes following long-term monocular deprivation (MD). (**A**) *Left*, schematic of mouse brain showing recording site in V1b, as well as sites of deprivation and inactivation in the contralateral and ipsilateral eyes, respectively. *Right*, timeline showing experimental manipulations and recording session times. (**B**) Longitudinal measurements of visual evoked potential (VEP) magnitude for stimulation of the contralateral eye at 0.2 cycles per degree (cpd) for three littermate treatment groups: *Sham*, sham contralateral eyelid suture P26-47 and fellow eye saline at P47; *MD*, contralateral eyelid suture P26-47 and fellow eye saline at P47; *MD then TTX*, contralateral eyelid suture P26-47 and fellow eye TTX at P47. Filled bars denote mean peak-to-peak magnitude and open circles denote individual biological replicates. Asterisk denotes a significant difference in VEP magnitude ($p < 0.05$) compared to before treatment (0 weeks). Analyses were performed using a two-way repeated measures ANOVA (treatment × time, $F_{(8,156)}=6.925$, $p < 0.0001$) followed by Dunnett's multiple comparisons tests (Sham, 0 vs. 1, 2, 3, 4 weeks: $p = 0.4963, 0.9774, 0.9986, 0.9999$; MD, 0 vs. 1, 2, 3, 4 weeks: $p = 0.5051, 0.5160, 0.9756, 0.9640$; MD then TTX, 0 vs. 1, 2, 3, 4 weeks: $p = 0.0052, 0.0024, 0.0007, 0.0010$). (**C**) Contralateral VEP magnitude over time for stimulation at 0.2 cpd, with Sham in black (dashed), MD in gray, and MD then TTX in blue. Open circles denote mean, error bars denote SEM; average waveforms shown above plot; gray and blue asterisks denote significant difference compared to Sham ($p < 0.05$) for the MD and MD then TTX groups, respectively, computed using Dunnett's multiple comparisons tests (MD vs. Sham at 1, 2, 3, 4, 5 weeks: $p = 0.0050, 0.0414, 0.0356, 0.0025, 0.0077$; MD then TTX vs. Sham at 1, 2, 3, 4, 5 weeks: $p = 0.0011, 0.9806, 0.6524, 0.5640, 0.8190$). (**D**) Contralateral VEP magnitude across different spatial frequencies just after opening deprived eye but before treatment (0 weeks) and 4 weeks after opening deprived eye and inactivating fellow eye. Symbols/colors same as C. Post hoc comparisons performed using Dunnett's multiple comparisons tests (0 weeks, Sham vs. MD at 0.05, 0.2, 0.4 cpd: $p = 0.0054, 0.0050, 0.0002$; 0 weeks, Sham vs. MD then TTX at 0.05, 0.2, 0.4 cpd: $p = 0.0006, 0.0011, < 0.0001$; 4 weeks, Sham vs. MD at 0.05, 0.2, 0.4 cpd: $p = 0.0311, 0.0077, 0.0157$; 4 weeks, Sham vs. MD then TTX at 0.05, 0.2, 0.4 cpd: $p = 0.9319, 0.8190, 0.6413$). (**E–G**) Same as B–D but for ipsilateral (fellow, inactivated) eye, with yellow denoting the MD then TTX condition in F–G. Analyses were performed using a two-way repeated measures ANOVA (treatment × time, $F_{(8,156)}=1.464$, $p = 0.1745$), with the absence of a significant interaction suggesting that there were no differences over time that could be attributed to

*Figure 2 continued on next page*

*Figure 2 continued*
treatment.
The online version of this article includes the following source data for figure 2:
**Source data 1.** VEP magnitudes in mouse V1 after long-term MD (or sham) followed by fellow eye inactivation (or saline).

pathogenesis of amblyopia. The electroencephalogram was recorded non-invasively over V1 during monocular viewing of phase-reversing grating stimuli at a range of spatial frequencies (*Figure 4A*). The response to visual stimulation was quantified by calculating the total power at the 2 Hz phase reversal frequency and the next six harmonics (*Figure 4B*). We compared these to the total power at control frequencies offset from the 2 Hz fundamental frequency and its harmonics by 0.45 Hz. As expected, the lowest spatial frequency presented (0.05 cycles per degree [cpd]) evoked the largest response. Using this approach, we were able to reliably detect responses to stimuli for spatial frequencies up to 0.5 cpd (*Figure 4C*) with response profiles similar to time-domain VEP analysis (*Figure 4—figure supplement 1*).

We tracked longitudinally the visually evoked responses in cats that had undergone 3 weeks of MD until P51, an age at which RO is minimally effective at reversing the effects of MD (*Blakemore and Van Sluyters, 1974*). After confirming the stable loss of visual responsiveness to the deprived eye, fellow eye inactivation was achieved over 8–10 days using intravitreal TTX (*Figure 4D–F*). As cats are precious, in these experiments we opted for a treatment duration of sufficient length to adequately test our hypothesis. Before MD, all cats had balanced responses to stimulation of either eye, and this was strongly shifted after MD (*Figure 4G*; *Figure 4—figure supplement 2*) due to weaker responses to deprived eye stimulation (*Figure 4E–F*). In three of the four animals, we allowed 1–2 weeks of binocular vision prior to initiating inactivation and verified that the impairment in deprived eye responses was stable. During the period of inactivation, fellow eye responses were reduced to the level of control frequencies and gray screen values, as expected (*Figure 4F*). Meanwhile, responses to stimulation of the previously deprived (non-inactivated) eye were gradually potentiated (*Figure 4E–F*). After the TTX wore off, responses to stimulation of the fellow eye returned, and we again observed balanced V1 responses to stimulation of both eyes that persisted for many weeks without sign of regression (*Figure 4E–G*; *Figure 4—figure supplement 2*). These observations were consistent across all four animals (*Figure 4G*) at spatial frequencies up to the 0.5 cpd detection limit (*Supplementary file 3*), and there was no discernable decrement in efficacy for the animal that underwent fellow eye inactivation at the oldest postnatal age (P65; *Figure 4E–F*; *Figure 4—figure supplement 2A*; *Supplementary file 3*). Post-mortem examination confirmed normal ocular histology in all animals, consistent with previous observations in the cat (*DiCostanzo et al., 2020*). These results demonstrate that, following amblyogenic rearing, fellow eye inactivation promotes a balanced recovery in cat V1 downstream of both eyes, and this intervention is efficacious at ages when previously established sensitive periods indicated limited potential for reversal. Although quantitative behavioral assessment was not performed, qualitative observation of the kittens showed unambiguously that vision and visually guided behavior was restored by treatment.

## Fellow eye inactivation corrects anatomical effects of MD in cat dLGN

In cats and primates, inputs from the two eyes are distributed to different laminae of the dorsal lateral geniculate nucleus (dLGN). In these highly visual species, an anatomical hallmark of MD is the reduction of dLGN neuron size within laminae downstream of the deprived eye (*Duffy and Slusar, 2009*; *Wiesel and Hubel, 1963*), a phenomenon associated with shrinkage of ocular dominance columns (*Hubel et al., 1977*) and dependent on synaptic modification in V1 (*Bear and Colman, 1990*). To explore the effect of fellow eye inactivation on an anatomical marker of ocular dominance plasticity, we compared post-mortem analysis of dLGN soma size from three cats that had undergone 3 weeks of MD followed by fellow eye inactivation with age-matched controls. Animals that underwent MD alone showed the classic shrinkage of dLGN cells in deprived laminae compared with the non-deprived lamina in the binocular segment (*Figure 4H–I*). In contrast, cats subjected to MD that additionally underwent fellow eye inactivation showed comparable soma sizes in deprived and non-deprived dLGN lamina (*Figure 4H–I*). Importantly, this was not the result of atrophy of cells downstream of both eyes, as the

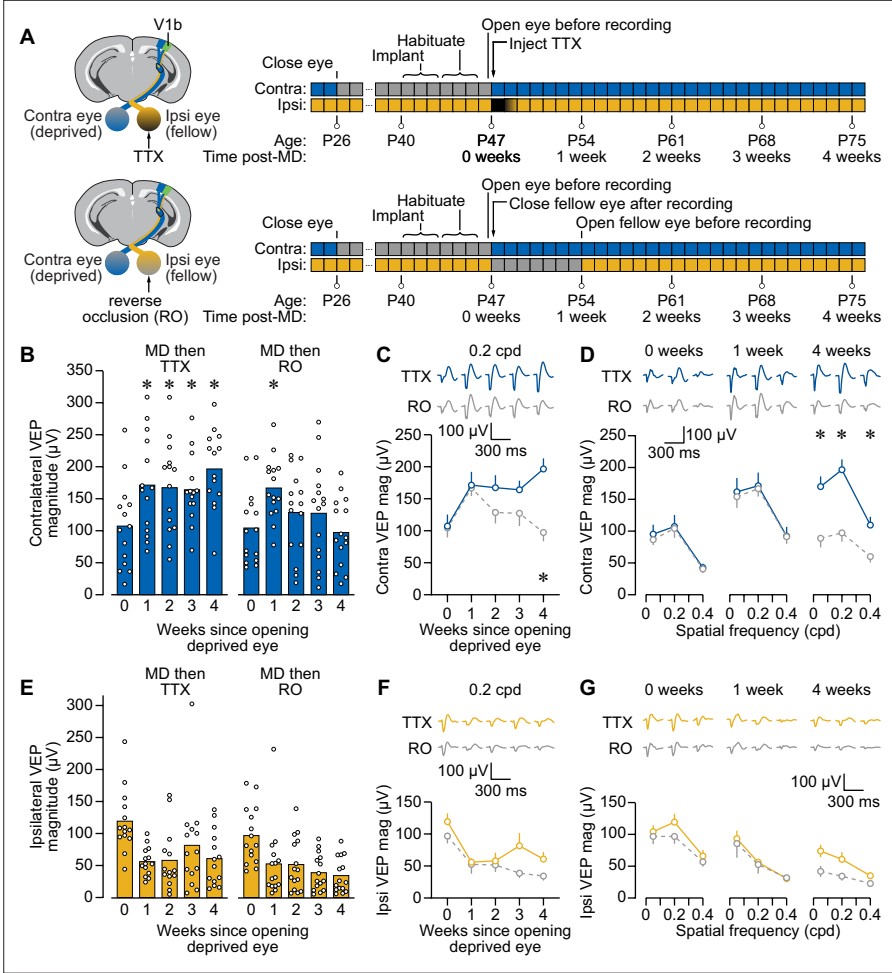

**Figure 3.** Reverse occlusion (RO) improves deprived eye vision following long-term monocular deprivation (MD), but the recovery is not lasting. (**A**) Schematic and timelines for experiment comparing fellow eye inactivation to RO following 3 weeks MD. (**B–G**) Same format as *Figure 2B–G*, but for two littermate treatment groups: *MD then TTX*, contralateral eyelid suture P26-47 and fellow eye TTX injection + sham suture at P47 + sham opening at P54; *MD then RO*, contralateral eyelid suture P26-47 and fellow eye saline + RO at P47-54. MD then TTX group is blue for C–D and yellow for F–G. MD then RO group is gray (dashed) for C–D and F–G. Except where otherwise noted, all data in this figure are for phase-reversing sinusoidal grating stimulation at a spatial frequency of 0.2 cycles per degree (cpd). Analyses in B and E were performed using two-way repeated measures ANOVA tests (contralateral eye, treatment × time, $F_{(4,108)}=6.363$, $p = 0.0001$; ipsilateral eye, treatment × time, $F_{(4,108)}=1.474$, $p = 0.2152$), with the significant interaction for the contralateral eye motivating Dunnett's multiple comparisons tests (MD then TTX, 0 vs. 1, 2, 3, 4 weeks: $p = 0.0057, 0.0065, 0.0437, 0.0042$; MD then RO, 0 vs. 1, 2, 3, 4 weeks: $p = 0.0035, 0.4119, 0.5738, 0.9831$). Dunnett's post hoc tests were also used for analyses in C (MD then TTX vs. MD then RO at 0 and 1 weeks: $p > 0.9999$, at 2, 3, 4 weeks: $p = 0.5361, 0.5446, 0.0004$) and D (0 weeks, MD then TTX vs. MD then RO at 0.05, 0.2, 0.4 cpd: $p = 0.9917, >0.9999, 0.9998$; 1 week, MD then TTX vs. MD then RO at 0.05, 0.2, 0.4 cpd: $p = 0.9995, >0.9999, >0.9999$; 4 weeks, MD then TTX vs. MD then RO at 0.05, 0.2, 0.4 cpd: $p = 0.0041, 0.0004, 0.0255$).

The online version of this article includes the following source data for figure 3:

**Source data 1.** VEP magnitudes in mouse V1 after long-term MD followed by fellow eye inactivation or reverse occlusion.

soma size of animals undergoing fellow eye inactivation was statistically indistinguishable from those of normally reared controls (*Figure 4J*). These results provide evidence that, in addition to physiological recovery from MD (*Figures 2B–G–4E–G*; *Figure 4—figure supplement 2*; *Supplementary file 3*), fellow eye inactivation corrects an anatomical correlate of ocular dominance plasticity in the cat visual pathway.

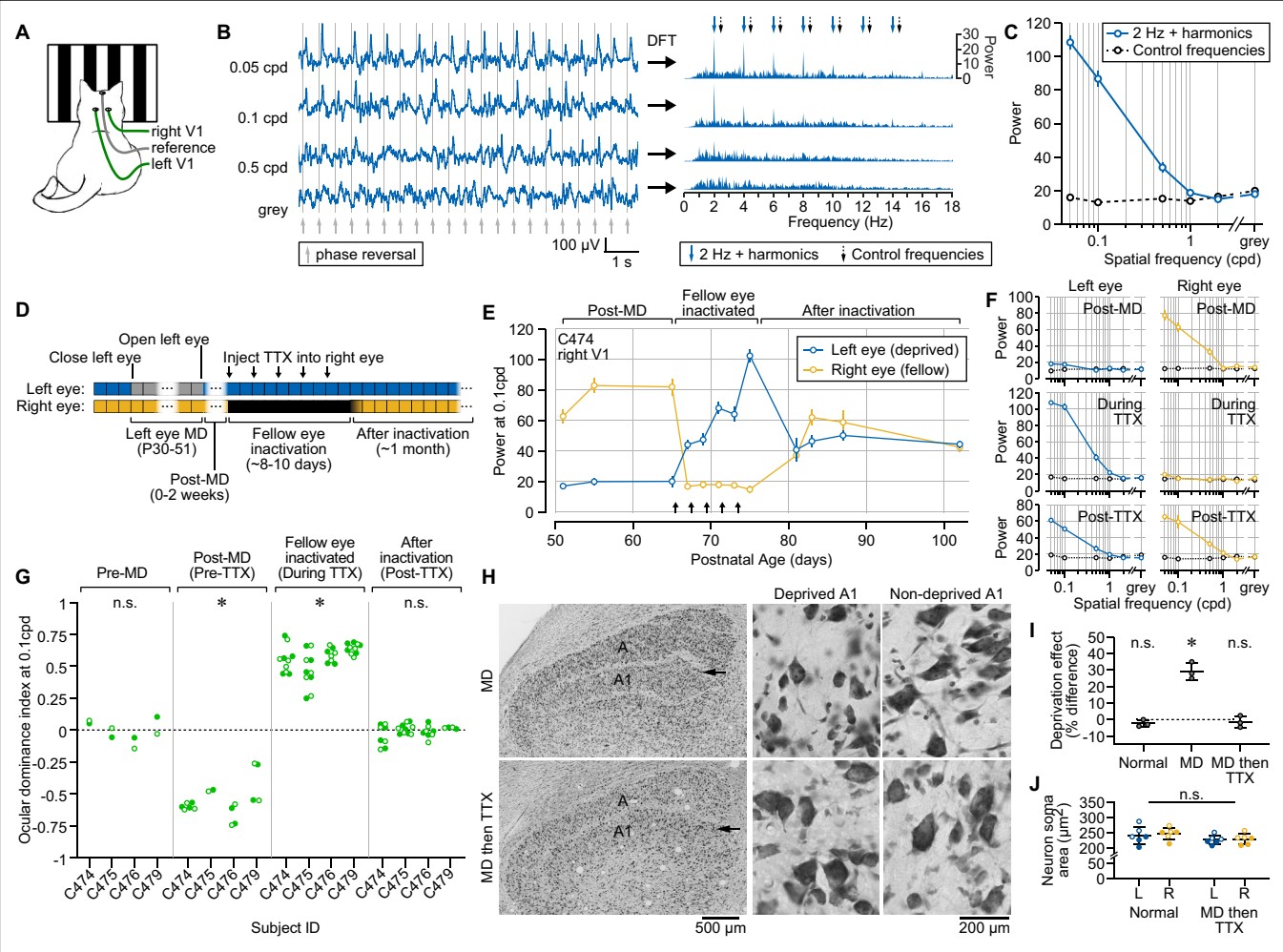

**Figure 4.** Fellow eye inactivation promotes stable functional and structural recovery in cats following long-term monocular deprivation (MD). (**A**) Recording and visual stimulation setup for measuring visually evoked responses non-invasively in cats. (**B**) Methodology for computing visually evoked responses from scalp surface field potential. *Left*, example raw field potential time series recorded from V1 during presentation of phase-reversing visual stimuli at a range of spatial frequencies. Gray lines denote timing of phase reversals. *Right*, data on left shown in the frequency domain following discrete Fourier transform (DFT). Blue arrows point to peaks in spectral power at 2 Hz (the phase reversal frequency) and six harmonics, representing visually evoked responses. Black arrows point to the frequencies used for control power measurements. (**C**) Total power at the visually driven (phase-reversing) frequency and its harmonics (blue) vs. control (black) across a range of spatial frequencies. Values computed from same recording shown in B. (**D**) Timeline showing experimental manipulations and recording session times. (**E**) Total power of visually evoked responses for one animal (C474, right primary visual cortex [V1]) viewing visual stimuli through the deprived left eye (blue) vs. the fellow right eye (yellow). Arrows denote time of tetrodotoxin (TTX) injections. Error bars, SEM. Spatial frequency, 0.1 cycles per degree (cpd). (**F**) Total power of visually driven responses for the same animal shown in E across a range of spatial frequencies before (top), during (middle), and after (bottom) inactivation of the fellow eye with TTX. Deprived left eye shown in blue, and fellow right eye shown in yellow. Black symbols denote control frequencies as in C. (**G**) Ocular dominance indices calculated for four cats before MD, after MD (but before inactivation), during inactivation, and after the fellow eye was no longer inactivated. Data are shown for both right (closed circles) and left (open circles) V1. Positive and negative values indicate response biases toward the left and right eyes, respectively, while the dashed line at 0 indicates balanced V1 responses expected from a neurotypical animal. As a reference, indices for the right hemisphere of C474 were calculated from data shown in E. Asterisks denote significant differences (p > 0.05) from the hypothesized value of 0, analyzed using Wilcoxon signed-rank tests (pre-MD, p = 0.9453; post-MD, p < 0.0001; fellow eye inactivated, p < 0.0001; post-TTX, p = 0.8236). (**H**) Left, low magnification image of the lateral geniculate nucleus (LGN) stained for Nissl substance after 3 weeks MD (top), or 3 weeks MD followed by fellow eye inactivation (bottom). Arrow indicates lamina A1, ipsilateral to the deprived eye. Middle and right, high magnification images from deprived (middle) and non-deprived (right) A1 layers for the same conditions shown on left. (**I**) Deprivation effect, stereological quantification of neuron soma size within deprived and non-deprived A and A1 layers, for normally reared cats, cats undergoing 3 weeks of MD, and cats undergoing 3 weeks of MD followed by fellow eye inactivation. Asterisk denotes significant difference (p > 0.05) from the hypothesized value of 0%, analyzed using a one-sample t-test (normal, p = 0.1917; MD, p = 0.0100; MD then TTX, p = 0.5716). (**J**) Average soma size for neurons in LGN layers downstream of the deprived left eye (L) or the fellow right eye (R) for normally reared animals compared to those undergoing 3 weeks of MD followed by fellow eye inactivation. There was no significant difference between the groups (Welch's ANOVA, W = 1.426, p = 0.2881). Open and closed symbols indicate values drawn from laminas A1 and A, respectively. Black lines

*Figure 4 continued on next page*

*Figure 4 continued*

indicate mean and SEM for each group.

The online version of this article includes the following source data and figure supplement(s) for figure 4:

**Source data 1.** VEP magnitudes from cat V1 after long-term MD followed by fellow eye inactivation.

**Figure supplement 1.** Time-domain visual evoked potentials (VEPs) from cat scalp surface field potential.

**Figure supplement 2.** Trajectory of deprivation-driven ocular dominance shift and inactivation-mediated recovery in individual cats.

## Discussion

In animals, long-term MD models the cause of the most severe form of human amblyopia, for which treatment options are absent or limited. Traditional therapies, such as patching the fellow eye, are beneficial only when initiated at very young ages (*Birch and Stager, 1996*). Even when patching is initially successful, recurrence of amblyopia is common (*Bhola et al., 2006*; *Holmes et al., 2004*). These observations supported the view that the brain loses plasticity shortly after birth, when synaptic connections are essentially set for life (*LeVay et al., 1980*). However, experiments over the past 40 years in a number of species have shown repeatedly that under certain conditions synaptic connections in V1 remain mutable seemingly across the lifespan. Notwithstanding the enormous potential these findings have for the treatment of amblyopia, efforts to translate this knowledge to human benefit have so far been unsuccessful. The steep challenge that remains is to devise a strategy for harnessing this plasticity to promote recovery of function that can be applied in a clinical setting.

The current study was motivated by numerous human clinical reports that stable and severe amblyopia can sometimes remit in adults when the fellow eye is damaged or removed (*El Mallah et al., 2000*; *Kaarniranta and Kontkanen, 2003*; *Klaeger-Manzanell et al., 1994*; *Rahi et al., 2002*; *Vereecken and Brabant, 1984*). Our experiments were designed to test the hypothesis that this remission is enabled, not by permanent removal of interocular suppression by the fellow eye, but by eliminating activity in the eye only for as long as required to allow connections from the amblyopic eye to become reestablished in V1. The data, obtained in two evolutionarily distant animal models, clearly indicate that stable and severe amblyopia can be reversed following temporary inactivation of the fellow eye. This recovery from amblyopia is durable, persisting for weeks after the TTX has worn off and vision is fully restored in the inactivated eye, and is observed in animals at ages beyond the classically defined critical period.

While our project was underway, we learned that the *Rd8* mutation of the *Crb1* gene is present in the C57BL/6 N vendor line used in many mouse studies, including our own (*Mattapallil et al., 2012*). This mutation causes late onset retinal degeneration detectable by multiple light-colored spots in the fundus of the eye that correspond histologically to retinal folds (*Chang et al., 2002*). As part of our routine post-mortem tissue evaluation, we examined retinas from both eyes under a microscope to check for abnormalities in every mouse. In total, data from three mice were excluded, each because of darkened patches or dimples observed in the peripheral retinas of eyes that had received intravitreal injections of TTX (n = 2) or saline (n = 1). We ascribed these lesions to poorly executed injections, as they were not observed in eyes that were untreated, but we cannot rule out the possibility that one or more of these abnormalities was the result of the *Rd8* mutation. Regardless of cause, every animal with an observable retinal abnormality was excluded from our analysis. Although there might have been lesions too small for us to detect, we are confident in the validity of our conclusions for several reasons. First, any retinal degeneration would be expected to affect both eyes equally and be present regardless of treatment. Second, the oldest mice we used were 3 months of age, whereas the retinal degeneration caused by the *Rd8* mutation is most severe in animals 10 months of age or older (*Aleman et al., 2011*; *Aredo et al., 2015*). Third, the *Rd8* mutation in C57BL/6 N mice does not alter VEPs elicited by full-field visual stimuli, electroretinograms, or retinal ganglion cell and optic nerve integrity (*Stojic et al., 2017*). Finally and perhaps most importantly, our conclusions generalize to the cat.

The current findings can be understood in the context of 'Hebbian' models of synaptic strengthening. By inactivating the dominant input, adaptations occur in V1 that enable visual experience to drive potentiation of synapses that are otherwise too weak to be modified (see, e.g., *Clothiaux et al., 1991*). In this conceptual framework, recovery might be further enhanced as activity in the fellow

eye returns, analogous to the phenomenon of associative long-term potentiation (*Barrionuevo and Brown, 1983*). We note that this additional bootstrapping of the weak eye input would not be available if the fellow eye were permanently removed, perhaps explaining why in humans, loss of the fellow eye leads to recovery from amblyopia in only ~10% of cases after age 11 (*Rahi et al., 2002*). We are far more successful in reversing amblyopia after the critical period in the animal models using temporary inactivation of the fellow eye than what has been observed using enucleation (*Dräger, 1978*; *Harwerth et al., 1984*; *Kratz and Lehmkuhle, 1983*).

The end of the critical period is typically defined as the age when RO is no longer able to reverse the effects of early MD. In mice, the critical period has been estimated to end at ~P32 (*Gordon and Stryker, 1996*), so we were surprised to see that 7 days of RO initiated at P47 restored responses through an eye that had been rendered amblyopic with 3 weeks of MD. However, unlike what was observed with fellow eye inactivation, the improvement after RO was short-lived. Similar reemergence of visual impairment following RO has been reported in amblyopic kittens (*Mitchell et al., 1984*). It is noteworthy that although occlusion therapy is the standard of care for visual correction in children, approximately one-quarter of patients experience worsening vision through the amblyopic eye within a year after treatment concludes (*Bhola et al., 2006*; *Holmes et al., 2004*). These clinical findings highlight the importance of evaluating the long-term stability of recovery in preclinical studies. While a number of potential therapies for amblyopia have been identified in rodent models, long-term stability is rarely monitored.

Occlusion and retinal inactivation are qualitatively different manipulations of visual experience, and a key scientific takeaway is that they have profoundly different long-term consequences on synaptic efficacy in the visual cortex. Eye patching and eyelid closure degrade image form, but have little effect on the overall activity of retinal ganglion cells (*Kuffler et al., 1957*). Rather, occlusion replaces spatiotemporally structured patterns of activity with stochastic noise. During early development, it has been shown that this retinal noise can rapidly trigger the synaptic depression in V1 that causes amblyopia (*Cooke and Bear, 2014*). In contrast, visual responses and thalamocortical synaptic strength are preserved in V1 following a comparable period of intravitreal TTX (*Coleman et al., 2010* ; *Frenkel and Bear, 2004*; *Iurilli et al., 2012*; *Rittenhouse et al., 1999*). Although RO can promote recovery during the critical period, our experiments show that only inactivation produces a durable effect when initiated later in life.

Previous studies in cats and rodents showed that recovery from a brief period of MD could be promoted by prolonged immersion in complete darkness as well as by temporary bilateral retinal inactivation (*Duffy and Mitchell, 2013*; *Fong et al., 2016*; *He et al., 2007*). It is possible that dark exposure, binocular TTX, and fellow eye inactivation all tap into a common mechanism, but there are some important differences. To be effective, the period of darkness must be long (≥10 days) and cannot be interrupted, even briefly, by light exposure (*Mitchell et al., 2016*). Treatment of both retinas with TTX overcomes some of these limitations, but so far has been shown only to reverse the effects of 1 week MD (*Fong et al., 2016*), which may be too brief to accurately model human deprivation amblyopia. Darkness and binocular TTX are also less effective than fellow eye inactivation in promoting anatomical recovery beyond the critical period peak (*Duffy et al., 2018*). Furthermore, from a clinical standpoint, total visual deprivation is not practical. These procedures are known to disrupt circadian rhythms and cause visual hallucinations (*Pang, 2016*), and would necessitate continuous patient care and supervision. Thus, finding that monocular TTX treatment is effective in reversing amblyopia caused by long-term MD in both cats and mice represents a substantial advance.

The precise mechanisms that enable connections downstream of the amblyopic eye to gain strength following fellow eye inactivation remain to be determined. However, consideration of the published consequences of monocular inactivation and related manipulations may provide some insight. Similar to our observation that retinal inactivation significantly potentiates responses through the non-inactivated eye, both monocular enucleation and optic nerve crush in adult rodents augment the spared eye responses (*Nys et al., 2014*; *Vasalauskaite et al., 2019*). Unilateral retinal inactivation and enucleation both reduce markers of synaptic inhibition in V1 of monkeys (*Hendry et al., 1990*; *Hendry et al., 1994*; *Hendry and Jones, 1988*) and mice (*Barnes et al., 2015*; *Maffei and Turrigiano, 2008*), but interestingly, not in cats (*Bear et al., 1985*; *Benson et al., 1989*). Our data argue against the hypothesis that stable recovery is accounted for by the *permanent* reduction in interocular suppression, as this is not observed after unilateral TTX in the neurotypical mice (*Figure 1D*) and

would not account for the anatomical recovery of dLGN soma size in the cat (*Figure 4J*). However, a change in inhibition, even if transient, could be sufficient to promote Hebbian potentiation of excitatory synapses serving the amblyopic eye. Excitatory synaptic plasticity could also be facilitated by increases in principal cell excitability (*Barnes et al., 2015*; *Maffei and Turrigiano, 2008*) and adjustments in NMDA receptor properties (*Philpot et al., 2003*; *Quinlan et al., 1999*) that have been observed after dark exposure, retinal inactivation, or enucleation. Any or all of these homeostatic adjustments after fellow eye inactivation could account for the observed recovery of vision after the critical period, and it will be of interest in future studies to pinpoint the essential modifications. Although we do not yet know the precise mechanism, this gap in knowledge does not preclude application of insights gained here to amblyopia treatment. The mechanism for recovery enabled by patch therapy during the critical period is still not known, and yet this is the current standard of care for amblyopia.

The strengths of the current study are that (1) it is one of the few to satisfy the 'two-species' rule for establishing potential clinical efficacy, (2) it distinguishes between two distinct hypotheses for how adult enucleation of the fellow eye enables recovery from amblyopia, and (3) the experiments demonstrate durable recovery from the effect of long-term MD when traditional treatment fails. An injection of TTX (or equivalent) into the eye is not without risks that need to be carefully considered before contemplating human application (see, e.g., *Dossarps et al., 2015*). However, histological analysis of the cat retina and optic nerve revealed no deleterious effect of 10 days of TTX treatment (*DiCostanzo et al., 2020*), and the electrophysiological analyses and behavioral observations in the current study indicate complete functional recovery of vision in the injected eye. Similarly, experiments in awake, behaving monkeys show full recovery of visual acuity and eye reflexes following intravitreal TTX (*Foeller and Tychsen, 2019*). Complete retinal inactivation in cats and monkeys can be achieved with 1.5–15 µg of TTX confined to the vitreous humor (*Ataman et al., 2016*; *DiCostanzo et al., 2020*). In humans, 30 µg of TTX administered subcutaneously twice a day for 4 days was reported to be safe and well tolerated (*Hagen et al., 2017*). In addition, new biodegradable polymers have recently been developed for therapeutic delivery of TTX to achieve prolonged and local sodium channel blockade without detectable systemic toxicity (*Zhao et al., 2019*). Thus, there may be a path forward to apply this strategy for the treatment of deprivation amblyopia in adult patients.

## Materials and methods
### Mouse studies
Mouse experiments were conducted using male and female wild-type animals on the C57BL/6 N background. Animals were purchased from Charles River (RRID:IMSR_CRL:27) for experimentation or as breeders for a colony maintained at MIT. Offspring were housed in groups of two to five same-sex littermates after weaning at P21 and maintained on a 12 hr light-12 hr dark cycle. All recordings were conducted during the light cycle. Food and water were available ad libitum. Rearing and experimental procedures were conducted in accordance with protocols approved by the Committee on Animal Care at MIT, and conformed to guidelines from the National Institutes of Health and the Association for Assessment and Accreditation of Laboratory Animal Care International.

### Experimental design
For all experiments, chronic electrode implant surgery was conducted between P40 and P44, and animals were habituated to head fixation and monocular gray screen viewing on two separate days between P42 and P46. For monocular inactivation experiments (*Figure 1*), baseline recordings were conducted followed by intravitreal TTX injections into the eye contralateral to the recording electrode. Recordings were then conducted 1 hr, 24 hr, 48 hr, 4 days, and 7 days after the injection. Target sample size (number of biological replicates) was computed in SPSS for a power of 0.85 and effect size measured in pilot data (not included in study). For MD experiments (*Figures 2 and 3*), eyelid suture (or sham suture) was performed at P26. Deprived eyes were re-opened (or sham re-opened) at P47 (5–7 days after implant surgery), and the first electrophysiological recording from V1 took place 45 min later. Shortly after the first recording session, animals underwent either intravitreal injections of TTX or saline into the fellow eye, in some experiments with RO or sham RO. Recording sessions occurred weekly thereafter until P75. All recordings were conducted blind to deprivation

and treatment conditions, and each littermate group was handled by the same experimenter each week. Target sample size was estimated based on previously published data (*Fong et al., 2016*). At the conclusion of the experiment, mice were euthanized, and the brains and eyeballs were harvested for post-mortem analyses. In order to be included in the final data set, mice needed to meet all of the following a priori determined inclusion criteria: mice maintained good health status throughout experiment; eyelid stayed fully closed during MD and RO; a visually evoked response greater than noise was detectable through at least one eye at baseline; cornea, retina, and all parts of eyeball were healthy and intact; an electrode track was observable within L4 of binocular V1; at least one other same-sex littermate was included in another experimental group. In total, 25 of 104 mice were excluded for failure to meet these criteria (sutured eyelid opened early, n = 5; poor signal-to-noise, n = 2; malocclusion, n = 3, corneal abrasion, n = 3; retinal abnormality, n = 3; electrode track location, n = 5; absence of same-sex littermate, n = 4). Exclusions were made blind to treatment group, except in the case where animals were removed due to absence of a same-sex littermate in another treatment group; this rule was applied only after all other blinded exclusions were performed. After unblinding, post hoc examination of the exclusions showed corneal abrasions only in MD eyes and retinal abnormalities only in the injected eyes, but otherwise no systematic bias in exclusions between groups. Sample sizes reported refer to biological replicates (distinct animals used to capture the biological variation associated with each condition). Individual animals were tracked longitudinally, although technical replicates (independent repeated measurements at each time point) were not used.

## Eyelid suture

Animals were anesthetized via inhalation of isoflurane (1–3% in oxygen), and body temperature was maintained on a heated surface at 37 °C. Both eyes were moistened with sterile saline. For MD and RO, the eye being sutured first had fur on upper and lower eyelid of trimmed, and the eyeball surface rinsed with sterile saline. A thin layer of ophthalmic ointment containing bacitracin, neomycin, and polymysin was placed on the eye. The eyelid was closed using a single mattress stitch of polypropylene suture (Ethicon, 7–0 Prolene, 8648 G), with care taken not to touch the corneal surface, and a square knot was tied on the exterior of the lower lid. Ophthalmic ointment was applied where the suture needle passed through the eyelid. Nails on the forepaws were filed or lightly trimmed. For sham sutures, the suture was cut and removed just prior to turning off anesthesia. To remove sutures (3 weeks later for MD and 1 week later for RO), mice were anesthetized and maintained as previously described. The suture was cut and removed, and eyelids were gently separated. The corneal surface was rinsed with sterile saline and examined for signs of damage. Sham animals were anesthetized, placed under the bright microscope for an equivalent amount of time as MD and RO animals, and had their sham MD or RO eyeball rinsed. Animals recovered from MD, RO, and eye opening procedures in their home cages.

## Intravitreal injections

Mice were anesthetized using isoflurane, maintained at 37 °C, and had both eyes moistened thereafter as described for eyelid sutures. For the injected eye, an ~500 nm incision was made at the temporal corner of the eye, and a sterile silk suture thread (Ethicon, 7–0 silk, 7733 G) was pulled through the exposed sclera. The suture was pulled toward the nasal aspect to secure the globe and expose the temporal aspect of the eyeball. A 30-gauge needle was used to penetrate to the sclera and globe. The eye was rinsed with sterile saline just prior to inserting a glass micropipette into the vitreous chamber. A nanoliter injector was used to deliver 1 μl of either TTX (1 mM in citrate buffer; abcam ab120055) or saline. The micropipette was removed 1 min after the injection. The eye was rinsed with sterile saline and the spot of penetration was coated with ophthalmic ointment containing bacitracin, neomycin, and polymysin. Animals recovered from intravitreal injection in their home cages.

## V1 implant surgery

Pre-operative buprenorphine (0.1 mg/kg s.c.) was administered, and mice were subsequently anesthetized (1–3% isoflurane) and maintained at 37 °C. Ophthalmic ointment was applied to both eyes (or on the eyelid surface for eyes that were sutured closed). The fur on top of the head was shaved and the exposed scalp was cleaned using ethanol (70% v/v) and a povidone-iodine solution (10% w/v). An incision was made along the midline and connective tissue on the skull surface was removed. A steel

post was affixed to the skull anterior to Bregma. A craniotomy was made over the right prefrontal cortex for implanting a silver wire reference electrode on the cortical surface. Another craniotomy was made over binocular V1 (3 mm lateral of Lambda) contralateral to the monocularly deprived (or sham MD) eye, and a tungsten microelectrode (300–500 kΩ; FHC 30070) was lowered to L4 (450 μm from cortical surface). The steel posts and male gold pins coupled to electrodes were secured to the skull using cyanoacrylate. The skull was then covered in dental cement (Lang Dental, Ortho-jet, 1530WHT, and 1306CLR). Animals were removed from anesthesia and transferred to a heated recovery chamber. Meloxicam (1 mg/kg s.c.) was administered during this initial recovery phase, as well as for 2 days post-operatively. Health was carefully monitored by experimenters and veterinary staff, and supplemental fluids or heat was delivered if needed.

## Electrophysiological recording

All mouse recordings were conducted in awake, head-fixed animals with full-field visual stimuli presented on an LCD monitor in the binocular visual field at a viewing distance of 20 cm. On the days prior to the initial recording, animals were habituated to head restraint, gray screen viewing, and an opaque occluder that restricted vision to one eye or the other. On each recording day, local field potential (LFP) data were recorded from layer 4 of binocular V1 during visual stimulation sessions lasting 13.5 min per eye. Continuous LFP data were collected, amplified, digitized, and low-pass filtered using the Recorder-64 system (Plexon). Other than the visual stimulus, the experimental room was kept dark and free of distractions (e.g., neither the experimenter nor other mice were present in the room during the recording session). Stimuli were generated using custom software written in MATLAB using the PsychToolbox extension (*Brainard, 1997*; *Pelli, 1997*) and consisted of sinusoidal oriented gratings phase reversing at 2 Hz. Stimuli were presented in blocks consisting of 50 phase reversals at 100 % contrast for each of the following pseudo-randomly presented spatial frequencies: 0.05, 0.2, 0.4 cpd. Each block of stimuli was separated by a 30 s presentation of a luminance-matched gray screen. A distinct orientation (offset by at 30° offset from any previously viewed orientation) was used each week, and only non-cardinal orientations were selected for presentation. All recordings were performed blind to treatment condition.

## Electrophysiological data analysis

The VEP was defined as the phase reversal-triggered LFP, averaged across all phase reversals within a recording session for each individual animal, time point, and viewing eye. VEP waveforms and amplitudes were extracted from recorded LFP data using software developed by Jeffrey Gavornik (github.com/jeffgavornik/VEPAnalysisSuite; *Gavornik, 2021*). VEP magnitude was defined as the peak-to-peak amplitude of each biphasic VEP waveform. Data for the intermediate spatial frequency (0.2 cpd) is used for visualization, but data for lower and higher spatial frequencies is provided in *Supplementary files 1-2*.

## Post-mortem analyses

Mice were deeply anesthetized using isoflurane and rapidly decapitated. Both eyeballs were removed for immediate ocular examination and dissection in phosphate-buffered saline (PBS). The corneas were examined under a microscope for signs of damage, and parts of the eye were thereafter dissected to look for abnormalities, particularly of the lens or retina. Meanwhile, the brains were harvested and fixed in 4 % paraformaldehyde at room temperature. Tissue remained in fixation solution for 72 hr and was thereafter stored in PBS. Coronal slices of the V1 were made at 50 μm using a vibrating microtome. Slices were rinsed with water and phosphate buffer, and then mounted on charged glass slides (Fisher Scientific, 12-550-15). The mounted slices dried at room temperature and ambient humidity for 24 hr. Nissl bodies were then stained using cresyl violet (VWR, 26089–20), and coverslipped with 1.5 glass (VWR, 48393–251) and touline-based mounting medium (Electron Microscopy Sciences, Permount, 17986–05). Slices were imaged on a confocal microscope (Olympus) using the transmitted light channel, and digital micrographs of electrode tracks were saved. Micrographs were compared to a mouse brain atlas to determine localization of electrode tracks in layer 4 of binocular V1. All analyses described were performed blind to treatment condition.

## Cat studies

Physiological and anatomical studies were conducted on 10 cats (6 male and 4 female) that were all born and raised in a closed breeding colony at Dalhousie University. Rearing and experimental procedures were conducted in accordance with protocols approved by the University Committee on Laboratory Animals at Dalhousie University, and conformed to guidelines from the Canadian Council on Animal Care.

### Experimental design

Animals were monocularly deprived for 3 weeks starting at the peak of the critical period (P30). Four animals subjected to physiological assessment had their deprived eye opened and were administered four to five intraocular injections of TTX into the fellow eye immediately (C475), or following 1 (C476, C479) or 2 weeks (C474) of binocular vision. Animals underwent electroencephalogram recordings at various points throughout the experimental timeline. The number of animals used in the study was determined in consultation with veterinary staff, with three littermates being tested first in parallel, and one animal being tested later to validate results in an age-matched animal from a different litter. Three animals (C474, C475, C476) were euthanized at ~P110, and tissue was harvested for anatomical analysis. Anatomical assessments were made on two additional groups of animals, which acted as our controls. The first control group (n = 3) was normally reared until age 14 weeks and then received five injections of vehicle solution (citrate buffer) into the right eye. The second control group (n = 3) was monocularly deprived at P30 for 3 weeks. Sample sizes of control groups for anatomy were designed to match the number of animals in the treatment group.

### Eyelid suture

MD was performed under general gaseous anesthesia (3–4% isoflurane in oxygen) and involved closure of the upper and lower palpebral conjunctivae of the left eye with sterile polyglactin 910 thread (Vicryl), followed by closure of the eyelids with silk suture. Following the procedure, animals were administered Metacam (0.05 mg/kg) for post-procedure analgesia, local anesthesia was produced with Alcaine sterile ophthalmic solution (1 % proparacaine hydrochloride; CDMV, Canada), and a broad-spectrum topical antibiotic (1 % Chloromycetin; CDMV) was administered to mitigate infection after surgery.

### Intravitreal injections

Upon completion of the 3 -week MD period, animals were anesthetized with 3–4% isoflurane and the eyelids were opened. Either immediately or after a period of binocular vision, animals had their fellow eye inactivated with intravitreal injection of TTX (ab120055; abcam, Eugene, OR) that was solubilized in citrate buffer at 3 mM. Dosage was 0.5 μl/100 g body weight but irrespective of weight, injection volume never exceeded 10 μl per injection. This approximate dosage blocks action potentials of affected cells without obstructing critical cellular functions such as fast axoplasmic transport (*Ochs and Hollingsworth, 1971*). Injections were administered through a puncture made with a disposable sterile needle that created a small hole in the sclera located at the pars plana. Using a surgical micro-scope, the measured volume of TTX solution was dispensed into the vitreous chamber with a sterilized Hamilton syringe (Hamilton Company, Reno, NV) fixed with a 30-gauge needle (point style 4) that was positioned through the original puncture and about 5–10 mm into the chamber angled away from the lens. The total volume of TTX was dispensed slowly, and when complete the needle was held in place for about a minute before it was retracted. Following intraocular injection, topical antibiotic (1 % Chloromycetin) and anesthetic (Alcaine) were applied to the eye to prevent post-injection complica-tions, and Metacam (0.05 mg/kg) was administered for post-procedure analgesia. Animals received four to five injections, one every 48 hr, and for each injection the original puncture site was used to avoid having to make another hole. During the period of inactivation, we employed basic assessments of visual behavior and measured VEPs to confirm inactivation. We verified the absence of a pupillary light reflex as well as the lack of visuomotor behaviors such as visual placing, visual startle, and the ability to track a moving laser spot. These assessments were made while vision in the non-injected eye was briefly occluded with an opaque contact lens.

## Electrophysiological recording

All cat recordings were conducted in anesthetized animals with full-field visual stimuli presented on an LCD monitor in the binocular visual field at a viewing distance of 70 cm. In preparation for each recording session, animals were anesthetized with 1–1.5% isoflurane, and supplemental sedation was provided with intramuscular acepromazine (0.06–0.1 mg/kg) and butorphanol (0.1–0.2 mg/kg). Hair on the head was trimmed and a disposable razor was used to shave parts of the scalp where recording sites were located, two positioned ~2–8 mm posterior and 1–4 mm lateral to interaural zero over the presumptive location of the left and right primary visual cortices, and another site over the midline of the frontal lobes that acted as a reference. Electrode sites were abraded with Nuprep EEG skin preparation gel (bio-medical, Ann Arbor, MI), and were then cleaned with alcohol pads. Reusable 10 mm gold cup Grass electrodes (FS-E5GH-48; bio-medical) were secured to each electrode site using Ten20 EEG conductive paste (bio-medical, Ann Arbor, MI) that was applied to the scalp. Impedance of the recording electrodes was measured in relation to the reference electrode to ensure values for each were below 5 kΩ. Electrophysiological signals were amplified and digitized with an Intan headstage (RHD2132; 20 kHz sampling frequency), then recorded using an Open Ephys acquisition board and GUI software (Open Ephys, Cambridge, MA) (*Siegle et al., 2017*). Stimuli were generated using custom software developed in MATLAB using the Psychophysics Toolbox extension (*Brainard, 1997*; *Pelli, 1997*), and consisted of full contrast square wave gratings with a 2 Hz contrast reversal frequency (*Bonds, 1984*; *Norcia et al., 2015*; *Pang and Bonds, 1991*). Blocks of grating stimuli at different spatial frequencies (0.05, 0.1, 0.5, 1, and 2 cpd) or a luminance-matched gray screen were presented in pseudo-random order for 20 s each, with the gray screen also displayed during a 2 s interstimulus interval. Each block was repeated at least six times. Each eye was tested in isolation by placing a black occluder in front of the other eye during recording. Eyes were kept open with small specula, and the eyes were frequently lubricated with hydrating drops. Recording sessions lasted about 1 hr and animal behavior was observed for at least an additional hour post-recording to ensure complete recovery.

## Electrophysiological data analysis

The raw electroencephalogram was imported to MATLAB where it was high-pass filtered above 1 Hz, then subjected to Fourier analysis (*Bach and Meigen, 1999*; *Norcia et al., 2015*). The magnitude of VEPs was calculated as the sum of power at the stimulus fundamental frequency plus six additional harmonics (2–14 Hz) (*DiCostanzo et al., 2020*). Baseline nonvisual activity was calculated as the sum of power at frequencies 0.2 Hz offset from the visual response (2.2–14.2 Hz). Parallel analysis of EEG data was performed using the EEGLab and ERPLab toolboxes for MATLAB (*Black et al., 2017*; *Delorme and Makeig, 2004*; *Lopez-Calderon and Luck, 2014*). The data was bandpass filtered, segmented, and normalized for extracting event-related potentials for time-domain VEP analysis. Response profiles were similar using frequency- and time-domain analyses (*Figure 4C*; *Figure 4— figure supplement 1B*), but variability in peak VEP latencies and polarities after MD and during inactivation made frequency domain analysis a more practical choice for longitudinal comparisons. Ocular dominance index was computed as $(L-R)/(L + R)$, where L is the mean summed power through the left (deprived) eye and R is the mean summed power through the right (fellow) eye. Data for the intermediate detectable spatial frequency (0.1 cpd) is used for visualization, but data for lower and higher spatial frequencies is provided *Supplementary file 3*.

## Histology

In preparation for histology, animals were euthanized with a lethal dose of sodium pentobarbital (pentobarbital sodium; 150 mg/kg) and shortly thereafter exsanguinated by transcardial perfusion with ~150 ml of PBS followed by an equal volume of PBS containing 4 % dissolved paraformaldehyde. Brain tissue was immediately extracted and the thalamus was dissected from the remainder of the brain in order to prepare the LGN for sectioning and histological processing. Tissue containing the LGN was cryoprotected and then cut coronally into 25 μm thick sections using a sliding microtome. A subset of sections was mounted onto glass slides and stained with a 1 % Nissl solution (ab246817; abcam, Eugene, OR), and were then coverslipped with mounting medium (Permount). To analyze cell size, the cross-sectional area of neuron somata within A and A1 layers of the left and right LGN was measured from Nissl-stained sections using the nucleator probe from a computerized stereology

system (newCAST; VisioPharm, Denmark). All measurements were performed using a BX-51 compound microscope with a 60 × oil-immersion objective (Olympus; Markham, Ottawa, Canada). Neurons were distinguished from glial cells using established selection criteria (*Wiesel and Hubel, 1963*) that included measurement of cells with dark cytoplasmic and nucleolar staining, and with light nuclear staining. Adherence to these criteria permitted avoidance of cell caps and inclusion only of neurons cut through the somal midline. Approximately 500–1000 neurons were measured from each animal. For each animal, an ocular dominance index was computed using the average soma size from each LGN layer: ([left A1 + right A]–[left A + right A1]/[left A1 + right A]), which indicated the percentage difference between eye-specific layers.

## Statistical analyses

Statistical analyses were performed using Prism (GraphPad). Normality testing was performed using the Shapiro-Wilk, and the outcome was used to determine whether to proceed with parametric or nonparametric tests. Variance of all data sets was also computed and used to determine whether to select a test that assumed equal variances or not. Comparisons to noise levels (to meet inclusion criteria) were performed using a paired t-test (parametric) or Wilcoxon matched-pairs signed-rank test (nonparametric). Comparisons to a hypothetical mean (e.g., for normalized data) were performed using one-sample t-tests (equal variances) or Welch's ANOVA (unequal variances). To analyze differences in means over time for a single group, a one-way repeated measures ANOVA with Geisser-Greenhouse correction was used. To analyze differences in means over time for multiple groups, a two-way repeated measures ANOVA with Geisser-Greenhouse correction was used. For both repeated measures ANOVA tests, post hoc were performed only in cases where there was a significant effect of time (one-way) or a significant interaction (two-way). Dunnett's tests were performed for multiple comparisons to a single baseline value, and Sidak's tests were performed for multiple comparisons between time points of interest. Significance level α was set at 0.05, and when necessary p-values were adjusted for multiple comparisons prior to comparing to α. All p-values are reported in figure legends.

## Acknowledgements

We thank Julia Deere, Lisandro Martin, Jocelyn Yao, Nathan Liang, and Kerlina Liu for technical assistance on mouse experiments; Nathan Crowder for experimental design and software development on cat physiology experiments; Braden Kamermans for software support and technical assistance on cat experiments; Jeffrey Gavornik for software development for mouse experiments; Arnold Heynen, Kiki Chu, Erin Hickey, Athene Wilson-Glover, and Jessica Buckey for research and administrative support; Victoria Mulloy, Victoria Donovan, and Chris Harvey-Clark for animal and veterinary support; Donald Mitchell and Lara Pierce for helpful discussions; and Eric Gaier for constructive comments on the manuscript. This work was supported by NIH K99 EY029326 to M-fF, CIHR 153333 to KRD, and NIH R01 EY029245 to MFB. Additional support was provided by the Arnold and Mabel Beckman Foundation and the JPB Foundation.

## Additional information

### Funding

| Funder | Grant reference number | Author |
|--------|------------------------|--------|
| National Eye Institute | R01 EY029245 | Mark F Bear |
| National Eye Institute | K99 EY029326 | Ming-fai Fong |
| Canadian Institutes of Health Research | 153333 | Kevin R Duffy |

The funders had no role in study design, data collection and interpretation, or the decision to submit the work for publication.

## Author contributions
Ming-fai Fong, Conceptualization, Formal analysis, Investigation, Methodology, Project administration, Visualization, Writing – original draft, Writing – review and editing; Kevin R Duffy, Conceptualization, Formal analysis, Funding acquisition, Investigation, Methodology, Project administration, Resources, Supervision, Visualization, Writing – review and editing; Madison P Leet, Christian T Candler, Investigation, Validation, Writing – review and editing; Mark F Bear, Conceptualization, Funding acquisition, Project administration, Resources, Supervision, Visualization, Writing – original draft, Writing – review and editing

## Author ORCIDs
Ming-fai Fong http://orcid.org/0000-0002-2336-4531
Kevin R Duffy http://orcid.org/0000-0003-3871-3287
Mark F Bear http://orcid.org/0000-0002-9903-2541

## Ethics
For mice, all experiments, handling, and rearing procedures were conducted in accordance with protocols approved by the Committee on Animal Care (CAC) at MIT (Protocol #0618-043-21), and conformed to guidelines from the National Institutes of Health and the Association for Assessment and Accreditation of Laboratory Animal Care International. For cats, all experiments, handling, and rearing procedures were conducted in accordance with protocols approved by the University Committee on Laboratory Animals (UCLA) at Dalhousie University (Protocol #20-038), and conformed to guidelines from the Canadian Council on Animal Care. In all cases, every effort was made to minimize suffering of research subjects, including the use of anesthetics and analgesics with any surgical procedures. Animal health and welfare was carefully monitored by both researchers and trained veterinary staff.

## Decision letter and Author response
Decision letter https://doi.org/10.7554/eLife.70023.sa1
Author response https://doi.org/10.7554/eLife.70023.sa2

---

# Additional files

## Supplementary files
• Supplementary file 1. Table of monocular responses in mouse primary visual cortex (V1) after long-term monocular deprivation (MD) (or sham) followed by fellow eye tetrodotoxin (TTX) (or saline) across spatial frequencies.

• Supplementary file 2. Table of monocular responses in mouse primary visual cortex (V1) after long-term monocular deprivation (MD) followed by fellow eye inactivation or reverse occlusion across spatial frequencies.

• Supplementary file 3. Table of ocular dominance indices in cat primary visual cortex (V1) after long-term monocular deprivation (MD) followed by fellow eye tetrodotoxin (TTX) across spatial frequencies.

• Transparent reporting form

## Data availability
All data generated or analysed during this study are included in the manuscript and supporting files.

---

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
