## [Decision Letter]

**Acceptance summary:**

This study of experimentally-induced amblyopia provides compelling evidence that monocular inactivation of the fellow (good) eye with tetrodotoxin supports long-lasting recovery from the effects of monocular deprivation, as measured by visual evoked potentials in primary visual cortex. This finding has clinical relevance, and provides support for the hypothesis that temporary relief from interocular suppression can open a window for plasticity of connections supporting signaling from the deprived eye.

**Decision letter after peer review:**

Thank you for submitting your article "Correction of amblyopia in cats and mice after the critical period" for consideration by *eLife*. Your article has been reviewed by 2 peer reviewers, and the evaluation has been overseen by a Reviewing Editor and Lu Chen as the Senior Editor. The reviewers have opted to remain anonymous.

Essential revisions:

The authors must make it clear what the main advance(s) are in the current work beyond their own previous work, particularly the Fong et al. 2016 PNAS paper, and the work of other labs. What is new and why it is important?

1) The cover letter asserts that "No prior findings by us or others would have necessarily predicted the results we obtained in the current study", but after close reading of the text and extensive discussion among the reviewers and reviewing editor, it was still not clear what is unexpected in the current work.

2) The current manuscript used 3 weeks MD, which is described as "long-term", whereas the authors' previous Fong et al. 2016 PNAS work used 1 week MD, which is described as "short-term." Is there something scientifically or clinically significant about 3 weeks vs 1 week of MD, or a reason based on previous work to expect a different outcome or different mechanism for MD of these two durations? The 1 week MD seemed to have long term effects on visual processing, which were then reversed by binocular TTX. Most labs deprive for less than 5 days for short-term MD, so the distinction between the 1 week protocol in the authors' PNAS paper, when compared to 3 weeks in this current manuscript is less distinct. Do the authors believe that the augmentation responses in normotypical mice (Figure 1) involve different mechanisms than those underlying augmentation in LTMD mice?

3) In the Discussion, the authors highlight the clinical importance of being able to use monocular retinal inactivation with TTX vs binocular inactivation. Is there also a scientific take-away, beyond the extensive previous work in the literature using monocular manipulations?

4) A point the authors highlight as a strength of the current study is that it " distinguishes between two competing hypotheses for how adult enucleation of the fellow eye enables recovery from amblyopia" (the two models being disinhibition vs plasticity), but the support for this claim was not clear. The main evidence for plasticity vs disinhibition would seem to be the "1 hour" time point (methods say earliest recording was 45 min), which is only tested in Figure 1, and which uses inactivation of the contralateral retina, as opposed to the ipsilateral retina in the rest of the figures, which test recovery from MD. Also, the "immediate" tests still seem to provide ample time for plasticity to occur-45min to 1 hour is what is typically used in slice experiments to measure LTP and LTD.

There seems to be some inconsistency in the authors' discussion of this issue: they argue in the discussion that the non-inactivated eye response to temporary TTX injection points to homeostatic mechanisms rather than interocular suppression. However, on page 6, 1st paragraph of the results regarding normotypical mice, they write: "These results demonstrate that removal of the influence of the dominant eye immediately augments responses through the non-dominant eye, consistent with interocular suppression." However, these results are similar to those noted for LTMD experiments with persistent augmentation after washout of TTX. Please explain this contradiction.

5) TTX was injected once in mice and 4-5 times in cats; a rationale should be provided for these choices. Does one injection not work in cats? Also, the Fong et al. 2016 study used 2 TTX binocular injections in cats; are more injections required for monocular injections?

6) C57BL6/N mice from Charles River are known to have retinal degeneration, (Mattapallil et al., 2012, Investigative Ophthalmology & Visual Science, May 2012, Vol. 53, No. 6 ). Please report whether your mice have retinal degeneration, as this could influence the interpretation of the results, particularly in experiments involving older mice.

*Reviewer #1 (Recommendations for the authors):*

Involvement (or not) of some of the known mechanisms of critical period plasticity, including, excitation-inhibition balance, chondroitin sulphate proteoglycans (CSPGs), myelin-derived factors (Nogo, MAG, OMgp etc) signaling, histone deacetylases (HDAC), and PirB, should be tested in mice. The results, positive or negative or mixed, would be helpful for understanding the mechanism underlying temporary inactivation induced recovery.

[Editors' note: further revisions were suggested prior to acceptance, as described below.]

Thank you for submitting your revised article "Correction of amblyopia in cats and mice after the critical period" for consideration by *eLife*. Your article has been re-evaluated by a Reviewing Editor, who has indicated below two additional points that should be addressed.

Essential revisions:

1) The concern about retinal degeneration in C57/BL6N mice should be addressed in the manuscript as well as in the point-by-point response.

2) Please indicate in the manuscript the number of mice excluded from the analysis because of darkened patches on the retina or failure to meet other inclusion criteria.

---

## [Author Response]

Essential revisions:The authors must make it clear what the main advance(s) are in the current work beyond their own previous work, particularly the Fong et al. 2016 PNAS paper, and the work of other labs. What is new and why it is important?1) The cover letter asserts that "No prior findings by us or others would have necessarily predicted the results we obtained in the current study", but after close reading of the text and extensive discussion among the reviewers and reviewing editor, it was still not clear what is unexpected in the current work.

We thank the reviewers for the helpful comments, and appreciate the opportunity to address the points that require clarification. The current study is the first to demonstrate that silencing activity in one eye can promote stable recovery from amblyopia. These findings were unexpected for three reasons. First and most obviously, selective and reversible manipulation of activity in the fellow (non-deprived) eye has been attempted in many prior studies by simply occluding this eye (reverse occlusion), but this is well known to be ineffective in promoting recovery after an early critical period (Birch and Stager, 1996; Blakemore and Van Sluyters, 1974). Thus, based on the existing literature, there was no reason to suspect that silencing the fellow eye temporarily with TTX would be so efficacious. Second, recovery from amblyopia in previous rodent studies, including our own, was a consequence of *global* manipulations of V1 physiology, for example, by prolonged immersion in complete darkness (He et al., 2007), silencing both retinas with TTX (Fong et al., 2016), pairing visual stimulation with exercise (Kaneko and Stryker, 2014), environmental enrichment (Sale et al., 2007; Tognini et al., 2012), food restriction (Spolidoro et al., 2011), transplanting interneurons (Davis et al., 2015), dissolving perineuronal nets (Lensjo et al., 2017; Pizzorusso et al., 2006), invasive (Bochner et al., 2014) and systemic drug treatments (Grieco et al., 2020; Maya Vetencourt et al., 2008; Sansevero et al., 2019; Silingardi et al., 2010), or by genetic manipulations (Hensch et al., 1998; Morishita et al., 2010; Stephany et al., 2018; Syken et al., 2006). Thus, it is both surprising and exciting that a manipulation as simple, selective, and temporary as monocular retinal silencing effectively reverses amblyopia caused by prolonged monocular deprivation after the critical period. Furthermore, the fact that this was observed in both cats and mice indicates this is not a phenomenon limited to rodents with poorly differentiated visual systems and limited binocular vision. Third, the recovery from amblyopia observed after enucleating the fellow eye in previous animal studies has been modest, and ascribed solely to the persistent loss of interocular suppression (Harwerth et al., 1984; Hendrickson et al., 1977; Hoffmann and Lippert, 1982; Kratz and Spear, 1976). If this interpretation were the explanation for inactivation of the fellow eye, one would expect that recovery of amblyopic eye responses would revert once the TTX had worn off, rather than the substantial and sustained recovery that we document in our study. Thus, one would not have predicted effectiveness of unilateral retinal inactivation in producing sustained improvements in responses to the amblyopic eye based on previous findings. We have revised our manuscript to clarify these important points.

2) The current manuscript used 3 weeks MD, which is described as "long-term", whereas the authors' previous Fong et al. 2016 PNAS work used 1 week MD, which is described as "short-term." Is there something scientifically or clinically significant about 3 weeks vs 1 week of MD, or a reason based on previous work to expect a different outcome or different mechanism for MD of these two durations? The 1 week MD seemed to have long term effects on visual processing, which were then reversed by binocular TTX. Most labs deprive for less than 5 days for short-term MD, so the distinction between the 1 week protocol in the authors' PNAS paper, when compared to 3 weeks in this current manuscript is less distinct. Do the authors believe that the augmentation responses in normotypical mice (Figure 1) involve different mechanisms than those underlying augmentation in LTMD mice?

The significance of 3 weeks of MD is that the duration continues beyond the end of the critical period, when we can be sure there is no opportunity for spontaneous recovery that can come with binocular visual experience following short duration MD in younger animals. Studies in cats have shown that functional ocular dominance shifts can occur with short duration MD by synaptic modification alone, without the gross anatomical changes (e.g., altered ocular dominance columns or modification of LGN neurons) that are hallmarks of long-term MD. In addition, molecules such as neurofilament that are linked to deprivation-induced neural modifications continue to change beyond 1 week of monocular deprivation (Kutcher and Duffy, 2007). We believe that if our findings are to be relevant to treatment of human amblyopia caused by months or even years of altered visual experience, it is important to push the duration of MD in the animals to match the consequences of chronic deprivation.

Our 2016 paper used 1 week of MD in both mice and cats because we had identified this as the minimum duration of MD that produced long-lasting electrophysiological and behavioral deficits, which were not ameliorated by binocular visual experience alone. We referred to 1 week as “short-term MD” in comparison to the current study, but we understand the confusion given that many labs (including ours) routinely use durations of MD of 1-5 days. We have replaced the term "short-term" with "1 week” to remove this ambiguity and now indicate in the discussion the rationale for using 3 week MD.

The persistent augmentation observed after unilateral retinal inactivation in neurotypical mice fades over days and therefore differs from the stable enhancement observed in the amblyopic eye. We now emphasize this finding in the results and include relevant comparisons in Figure 1. However, we do believe the effect of silencing the dominant eye observed in the neurotypical mice is an important clue for how temporary retinal inactivation can open a window of opportunity for recovery from amblyopia. For example, if silencing one eye reduces excitatory drive onto inhibitory interneurons, this would lower the threshold for long-term potentiation of inputs from the amblyopic eye. A similar augmentation of the ipsilateral eye response is observed after longer term MD of the contralateral eye, and depends upon activation of NMDA receptors in both juvenile and adult mice (Cho et al., 2009; Sato and Stryker, 2008; Sawtell et al., 2003). As we discuss in the manuscript, this is one potential mechanism that we are eager to examine in future studies.

3) In the Discussion, the authors highlight the clinical importance of being able to use monocular retinal inactivation with TTX vs binocular inactivation. Is there also a scientific take-away, beyond the extensive previous work in the literature using monocular manipulations?

The take-away from extensive previous work was that monocular manipulations alone do not work after the critical period. The scientific take-away from the current work is that this dogma is incorrect, and that there is an important difference between degrading images in one eye (leaving spontaneous ganglion cell activity intact), and silencing ganglion cell activity altogether. Indeed, suturing closed one eye during the critical period invariably weakens cortical responses to this eye in neurotypical mice. Monocular retinal inactivation for a comparable period does not have this deleterious effect (Coleman et al., 2010; Frenkel and Bear, 2004; Iurilli et al., 2012; Rittenhouse et al., 1999), and the current study shows no detriment to the inactivated eye when treatment is applied beyond the critical period peak. We expanded the Discussion to emphasize this key point.

Without conducting additional studies in which rearing conditions and treatment parameters are perfectly matched, it is difficult to state definitively that monocular manipulations are superior to binocular (or other) manipulations. However, we have observed that anatomical recovery following long-term MD is superior with monocular inactivation than it is with either dark exposure or binocular retinal inactivation (Duffy et al., 2018). The apparent superiority of monocular inactivation in mature animals presumably originates from the activity driven by weak eye inputs during the period of inactivation. Unlike with binocular inactivation, weak-eye inputs continue to propagate unobstructed visually-driven activity that may be more conducive to recovery.

4) A point the authors highlight as a strength of the current study is that it " distinguishes between two competing hypotheses for how adult enucleation of the fellow eye enables recovery from amblyopia" (the two models being disinhibition vs plasticity), but the support for this claim was not clear. The main evidence for plasticity vs disinhibition would seem to be the "1 hour" time point (methods say earliest recording was 45 min), which is only tested in Figure 1, and which uses inactivation of the contralateral retina, as opposed to the ipsilateral retina in the rest of the figures, which test recovery from MD. Also, the "immediate" tests still seem to provide ample time for plasticity to occur-45min to 1 hour is what is typically used in slice experiments to measure LTP and LTD.There seems to be some inconsistency in the authors' discussion of this issue: they argue in the discussion that the non-inactivated eye response to temporary TTX injection points to homeostatic mechanisms rather than interocular suppression. However, on page 6, 1st paragraph of the results regarding normotypical mice, they write: "These results demonstrate that removal of the influence of the dominant eye immediately augments responses through the non-dominant eye, consistent with interocular suppression." However, these results are similar to those noted for LTMD experiments with persistent augmentation after washout of TTX. Please explain this contradiction.

We understand the confusion on this point, and appreciate the opportunity to clarify. Interocular suppression is an ongoing process that accounts for binocular rivalry—why different images presented to the two eyes at the same time are perceived sequentially rather than simultaneously (Mentch et al., 2019). Relief from interocular suppression occurs immediately and reversibly when one eye is occluded. Thus, it was not surprising to see the non-dominant eye responses increase as the activity was blocked in the dominant eye of neurotypical mice. However, it was unexpected that the enhancement of non-dominant eye responses lasted for days after the TTX had worn off and responses in the dominant eye had fully recovered. The explanation we favor is that temporary relief from interocular suppression can open a window for plasticity of excitatory connections serving the other eye. Thus, in the case of amblyopia, when the fellow (dominant) eye is inactivated, activity in the amblyopic eye is capable of driving the Hebbian plasticity in visual cortex that is the physical basis for the permanent recovery of vision. The competing idea is that there is permanent reduction in interocular suppression after temporary silencing of the fellow eye. This explanation is not consistent with our observations that in addition to electrophysiological responses, the soma sizes of dLGN neurons serving the amblyopic eye also recover in the weeks after temporary retinal silencing. The shrinkage of dLGN neurons after MD is a well-known structural correlate of weakened excitatory synaptic transmission in the visual cortex (Bear and Colman, 1990; Hubel et al., 1977), and their regrowth suggests a long-term recovery of the strength of these inputs. We have updated the text in the Abstract, Introduction, and Discussion to clarify the distinctions between these hypotheses.

5) TTX was injected once in mice and 4-5 times in cats; a rationale should be provided for these choices. Does one injection not work in cats? Also, the Fong et al. 2016 study used 2 TTX binocular injections in cats; are more injections required for monocular injections?

This remains to be investigated fully. As cats are precious, we opted for a treatment duration of sufficient length that our hypothesis could be adequately tested. However, as shown in Figure 4E, changes in deprived-eye responses are clearly evident after a single injection of TTX into the fellow eye.

6) C57BL6/N mice from Charles River are known to have retinal degeneration, (Mattapallil et al., 2012, Investigative Ophthalmology & Visual Science, May 2012, Vol. 53, No. 6 ). Please report whether your mice have retinal degeneration, as this could influence the interpretation of the results, particularly in experiments involving older mice.

This is a good point and is broadly relevant to vision researchers. Admittedly, we learned of the rd8 mutation in the Crb1 gene (present in the retinal Muller glial cells) of C57BL/6N mice in the middle of our experiments and elected to continue using this sub-strain for within-study consistency. As part of our routine post-mortem tissue evaluation, we examined retinas from both eyes under a microscope to check for abnormalities. In total, data from 3 mice were excluded, each because of darkened patches/dimples observed in the peripheral retina. We believe these were the result of a poorly-aligned intravitreal injection because unblinding revealed that all cases were in the injected eyes of mice administered TTX (n=2) or saline (n=1). Although we cannot rule out the possibility that one or more of these abnormalities was the result of the rd8 mutation, we can say any animal with an observable retinal abnormality was excluded regardless of cause. Although we acknowledge that longitudinal in-vivo fundus imaging or OCT would provide a stronger evaluation of smaller retinal lesions, we did not have these tools available. While conceding the possibility that there might be lesions too small for us to detect, we believe the overall interpretation of our data is valid for several reasons:

1. The C57BL/6N sub-strain was used throughout the study, and the age of animals at the time of monocular deprivation and intravitreal injection is the same for all animals, including littermate controls.

2. Both eyes are equally susceptible to retinal degeneration.

3. We used full-field visual stimuli, rather than stimulating an isolated portion of the retina.

4. The rd8 mutation is most severe in animals 10 months or older (Aleman et al., 2011; Aredo et al., 2015). Our oldest mice were 3 months old.

5. The rd8 mutation found in C57BL/6N mice does not alter visually-evoked potentials, electroretinograms, or retinal ganglion cell / optic nerve integrity (Stojic et al., 2017).

6. The findings generalize across species.

To avoid concerns about retinopathy in the future, newer experiments are being conducted on the C57BL/6J substrain. We have detected no difference in response to MD or TTX, but we will report if any differences emerge.

Reviewer #1 (Recommendations for the authors):Involvement (or not) of some of the known mechanisms of critical period plasticity, including, excitation-inhibition balance, chondroitin sulphate proteoglycans (CSPGs), myelin-derived factors (Nogo, MAG, OMgp etc) signaling, histone deacetylases (HDAC), and PirB, should be tested in mice. The results, positive or negative or mixed, would be helpful for understanding the mechanism underlying temporary inactivation induced recovery.

We enthusiastically agree with the reviewer that it is now of great interest to examine the potential contribution of each of these mechanisms to the effects we observe with temporary retinal silencing, and also to determine if these apply to species other than rodents. However, testing the contributions of each of these mechanisms will require a large-scale effort that is beyond the scope of the current project. As we note in the manuscript, the failure to have identified the mechanism for patch therapy has not precluded its use in clinical practice, so we believe our findings are of great value even without knowledge of the mechanism. Nevertheless, we are committed to investigating the mechanisms involved in promoting recovery in future studies.

References

Aleman, T.S., Cideciyan, A.V., Aguirre, G.K., Huang, W.C., Mullins, C.L., Roman, A.J., Sumaroka, A., Olivares, M.B., Tsai, F.F., Schwartz, S.B.*, et al.* (2011). Human CRB1-associated retinal degeneration: comparison with the rd8 Crb1-mutant mouse model. Invest Ophthalmol Vis Sci *52*, 6898-6910.

Aredo, B., Zhang, K., Chen, X., Wang, C.X., Li, T., and Ufret-Vincenty, R.L. (2015). Differences in the distribution, phenotype and gene expression of subretinal microglia/macrophages in C57BL/6N (Crb1 rd8/rd8) versus C57BL6/J (Crb1 wt/wt) mice. J Neuroinflammation *12*, 6.

Bear, M.F., and Colman, H. (1990). Binocular competition in the control of geniculate cell size depends upon visual cortical N-methyl-D-aspartate receptor activation. Proceedings of the National Academy of Sciences of the United States of America *87*, 9246-9249.

Birch, E.E., and Stager, D.R. (1996). The critical period for surgical treatment of dense congenital unilateral cataract. Invest Ophthalmol Vis Sci *37*, 1532-1538.

Blakemore, C., and Van Sluyters, R.C. (1974). Reversal of the physiological effects of monocular deprivation in kittens: further evidence for a sensitive period. The Journal of physiology *237*, 195-216.

Bochner, D.N., Sapp, R.W., Adelson, J.D., Zhang, S., Lee, H., Djurisic, M., Syken, J., Dan, Y., and Shatz, C.J. (2014). Blocking PirB up-regulates spines and functional synapses to unlock visual cortical plasticity and facilitate recovery from amblyopia. Sci Transl Med *6*, 258ra140.

Cho, K.K., Khibnik, L., Philpot, B.D., and Bear, M.F. (2009). The ratio of NR2A/B NMDA receptor subunits determines the qualities of ocular dominance plasticity in visual cortex. Proceedings of the National Academy of Sciences of the United States of America *106*, 5377-5382.

Coleman, J.E., Nahmani, M., Gavornik, J.P., Haslinger, R., Heynen, A.J., Erisir, A., and Bear, M.F. (2010). Rapid structural remodeling of thalamocortical synapses parallels experience-dependent functional plasticity in mouse primary visual cortex. J Neurosci *30*, 9670-9682.

Davis, M.F., Figueroa Velez, D.X., Guevarra, R.P., Yang, M.C., Habeeb, M., Carathedathu, M.C., and Gandhi, S.P. (2015). Inhibitory Neuron Transplantation into Adult Visual Cortex Creates a New Critical Period that Rescues Impaired Vision. Neuron *86*, 1055-1066.

Duffy, K.R., Fong, M.F., Mitchell, D.E., and Bear, M.F. (2018). Recovery from the anatomical effects of long-term monocular deprivation in cat lateral geniculate nucleus. J Comp Neurol *526*, 310-323.

Fong, M.F., Mitchell, D.E., Duffy, K.R., and Bear, M.F. (2016). Rapid recovery from the effects of early monocular deprivation is enabled by temporary inactivation of the retinas. Proceedings of the National Academy of Sciences of the United States of America *113*, 14139-14144.

Frenkel, M.Y., and Bear, M.F. (2004). How monocular deprivation shifts ocular dominance in visual cortex of young mice. Neuron *44*, 917-923.

Grieco, S.F., Qiao, X., Zheng, X., Liu, Y., Chen, L., Zhang, H., Yu, Z., Gavornik, J.P., Lai, C., Gandhi, S.P.*, et al.* (2020). Subanesthetic Ketamine Reactivates Adult Cortical Plasticity to Restore Vision from Amblyopia. Curr Biol *30*, 3591-3603 e3598.

Harwerth, R.S., Smith, E.L., 3rd, Crawford, M.L., and von Noorden, G.K. (1984). Effects of enucleation of the nondeprived eye on stimulus deprivation amblyopia in monkeys. Invest Ophthalmol Vis Sci *25*, 10-18.

He, H.Y., Ray, B., Dennis, K., and Quinlan, E.M. (2007). Experience-dependent recovery of vision following chronic deprivation amblyopia. Nature neuroscience *10*, 1134-1136.

Hendrickson, A., Boles, J., and McLean, E.B. (1977). Visual acuity and behavior of monocularly deprived monkeys after retinal lesions. Invest Ophthalmol Vis Sci *16*, 469-473.

Hensch, T.K., Fagiolini, M., Mataga, N., Stryker, M.P., Baekkeskov, S., and Kash, S.F. (1998). Local GABA circuit control of experience-dependent plasticity in developing visual cortex. Science *282*, 1504-1508.

Hoffmann, K.P., and Lippert, P. (1982). Recovery of vision with the deprived eye after the loss of the non-deprived eye in cats. Hum Neurobiol *1*, 45-48.

Hubel, D.H., Wiesel, T.N., and LeVay, S. (1977). Plasticity of ocular dominance columns in monkey striate cortex. Philos Trans R Soc Lond B Biol Sci *278*, 377-409.

Iurilli, G., Benfenati, F., and Medini, P. (2012). Loss of visually driven synaptic responses in layer 4 regular-spiking neurons of rat visual cortex in absence of competing inputs. Cereb Cortex *22*, 2171-2181.

Kaneko, M., and Stryker, M.P. (2014). Sensory experience during locomotion promotes recovery of function in adult visual cortex. *ELife 3*, e02798.

Kratz, K.E., and Spear, P.D. (1976). Postcritical-period reversal of effects of monocular deprivation on striate cortex cells in the cat. Journal of neurophysiology *39*, 501-511.

Kutcher, M.R., and Duffy, K.R. (2007). Cytoskeleton alteration correlates with gross structural plasticity in the cat lateral geniculate nucleus. Vis Neurosci *24*, 775-785.

Lensjo, K.K., Lepperod, M.E., Dick, G., Hafting, T., and Fyhn, M. (2017). Removal of Perineuronal Nets Unlocks Juvenile Plasticity Through Network Mechanisms of Decreased Inhibition and Increased Γ Activity. J Neurosci *37*, 1269-1283.

Maya Vetencourt, J.F., Sale, A., Viegi, A., Baroncelli, L., De Pasquale, R., O'Leary, O.F., Castren, E., and Maffei, L. (2008). The antidepressant fluoxetine restores plasticity in the adult visual cortex. Science *320*, 385-388.

Mentch, J., Spiegel, A., Ricciardi, C., and Robertson, C.E. (2019). GABAergic Inhibition Gates Perceptual Awareness During Binocular Rivalry. J Neurosci *39*, 8398-8407.

Morishita, H., Miwa, J.M., Heintz, N., and Hensch, T.K. (2010). Lynx1, a cholinergic brake, limits plasticity in adult visual cortex. Science *330*, 1238-1240.

Pizzorusso, T., Medini, P., Landi, S., Baldini, S., Berardi, N., and Maffei, L. (2006). Structural and functional recovery from early monocular deprivation in adult rats. Proceedings of the National Academy of Sciences of the United States of America *103*, 8517-8522.

Rittenhouse, C.D., Shouval, H.Z., Paradiso, M.A., and Bear, M.F. (1999). Monocular deprivation induces homosynaptic long-term depression in visual cortex. Nature *397*, 347-350.

Sale, A., Maya Vetencourt, J.F., Medini, P., Cenni, M.C., Baroncelli, L., De Pasquale, R., and Maffei, L. (2007). Environmental enrichment in adulthood promotes amblyopia recovery through a reduction of intracortical inhibition. Nature neuroscience *10*, 679-681.

Sansevero, G., Baroncelli, L., Scali, M., and Sale, A. (2019). Intranasal BDNF administration promotes visual function recovery in adult amblyopic rats. Neuropharmacology *145*, 114-122.

Sato, M., and Stryker, M.P. (2008). Distinctive features of adult ocular dominance plasticity. J Neurosci *28*, 10278-10286.

Sawtell, N.B., Frenkel, M.Y., Philpot, B.D., Nakazawa, K., Tonegawa, S., and Bear, M.F. (2003). NMDA receptor-dependent ocular dominance plasticity in adult visual cortex. Neuron *38*, 977-985.

Silingardi, D., Scali, M., Belluomini, G., and Pizzorusso, T. (2010). Epigenetic treatments of adult rats promote recovery from visual acuity deficits induced by long-term monocular deprivation. Eur J Neurosci *31*, 2185-2192.

Spolidoro, M., Baroncelli, L., Putignano, E., Maya-Vetencourt, J.F., Viegi, A., and Maffei, L. (2011). Food restriction enhances visual cortex plasticity in adulthood. Nat Commun *2*, 320.

Stephany, C.E., Ma, X., Dorton, H.M., Wu, J., Solomon, A.M., Frantz, M.G., Qiu, S., and McGee, A.W. (2018). Distinct Circuits for Recovery of Eye Dominance and Acuity in Murine Amblyopia. Curr Biol *28*, 1914-1923 e1915.

Stojic, A., Fairless, R., Beck, S.C., Sothilingam, V., Weissgerber, P., Wissenbach, U., Gimmy, V., Seeliger, M.W., Flockerzi, V., Diem, R.*, et al.* (2017). Murine Autoimmune Optic Neuritis Is Not Phenotypically Altered by the Retinal Degeneration 8 Mutation. Invest Ophthalmol Vis Sci *58*, 318-328.

Syken, J., Grandpre, T., Kanold, P.O., and Shatz, C.J. (2006). PirB restricts ocular-dominance plasticity in visual cortex. Science *313*, 1795-1800.

Tognini, P., Manno, I., Bonaccorsi, J., Cenni, M.C., Sale, A., and Maffei, L. (2012). Environmental enrichment promotes plasticity and visual acuity recovery in adult monocular amblyopic rats. PloS one *7*, e34815.

[Editors' note: further revisions were suggested prior to acceptance, as described below.]

Essential revisions:1) The concern about retinal degeneration in C57/BL6N mice should be addressed in the manuscript as well as in the point-by-point response.

We added a paragraph to the discussion.

2) Please indicate in the manuscript the number of mice excluded from the analysis because of darkened patches on the retina or failure to meet other inclusion criteria.

These are now indicated in the Methods section.